# De Novo Structural Determination of the Oligosaccharide Structure of Hemocyanins from Molluscs

**DOI:** 10.3390/biom10111470

**Published:** 2020-10-22

**Authors:** Pavlina Dolashka, Asya Daskalova, Aleksandar Dolashki, Wolfgang Voelter

**Affiliations:** 1Institute of Organic Chemistry with Centre of Phytochemistry, Bulgarian Academy of Sciences, 1113 Sofia, Bulgaria or adaskalova@orgchm.bas.bg (A.D.); adolashki@orgchm.bas.bg (A.D.); 2Interfacultary Institute of Biochemistry, University of Tuebingen, 72074 Tuebingen, Germany; wolfgang.voelter@uni-tuebingen.de

**Keywords:** hemocyanins, molluscs, mass spectrometric analysis

## Abstract

A number of studies have shown that glycosylation of proteins plays diverse functions in the lives of organisms, has crucial biological and physiological roles in pathogen–host interactions, and is involved in a large number of biological events in the immune system, and in virus and bacteria recognition. The large amount of scientific interest in glycoproteins of molluscan hemocyanins is due not only to their complex quaternary structures, but also to the great diversity of their oligosaccharide structures with a high carbohydrate content (2–9%). This great variety is due to their specific monosaccharide composition and different side chain composition. The determination of glycans and glycopeptides was performed with the most commonly used methods for the analysis of biomolecules, including peptides and proteins, including Matrix Assisted Laser Desorption/Ionisation–Time of Flight (MALDI-TOF-TOF), Liquid Chromatography - Electrospray Ionization-Mass Spectrometry (LC/ESI-MS), Liquid Chromatography (LC-Q-trap-MS/MS) or Nano- Electrospray Ionization-Mass Spectrometry (nano-ESI-MS) and others. The molluscan hemocyanins have complex carbohydrate structures with predominant *N*-linked glycans. Of interest are identified structures with methylated hexoses and xyloses arranged at different positions in the carbohydrate moieties of molluscan hemocyanins. Novel acidic glycan structures with specific glycosylation positions, e.g., hemocyanins that enable a deeper insight into the glycosylation process, were observed in *Rapana venosa*, *Helix lucorum*, and *Haliotis tuberculata*. Recent studies demonstrate that glycosylation plays a crucial physiological role in the immunostimulatory and therapeutic effect of glycoproteins. The remarkable diversity of hemocyanin glycan content is an important feature of their immune function and provides a new concept in the antibody–antigen interaction through clustered carbohydrate epitopes.

## 1. Introduction

Glycoproteins are of great importance for the optimal and proper functioning of many bioactivities in the human body. Absence or alteration of glycoproteins can cause a wide range of diseases, e.g., the glycated form of haemoglobin is responsible for the onset of diabetes [1]. Glycans also play an important role in cardiovascular disease, microbial and viral pathogenesis, development of tumors, tissue repair, strengthening of the immune system, and others [2,3,4]. As demonstrated by various studies [5,6], hemocyanins have gained a large amount scientific interest due to their interesting glycoprotein structures.

Hemocyanins are type-3 copper-binding proteins found freely dissolved in the hemolymph in two main phyla (mollusca and arthropoda), which have the same physiological function, but show significant differences in their structural organization and carbohydrate content. 

Molluscan hemocyanins are among the largest known glycoproteins with huge cylindrical multimeric forms and molecular masses ranging from 3.3 to 13.5 MDa. They form decamers or multi-decamers of 330- to 550-kDa subunits organised by more than seven different functional units (FUs) with molecular masses ranging from 45 to 65 kDa. Structural subunits assemble to di-decamers as presented for keyhole limpet hemocyanin (KLH1) in (Figure 1) [7].

Their structure has been investigated using a combination of electron cryo-microsopy and X-ray crystallography (Figure 2). The represented X-ray crystal structure of the intact 3.8-MDa molecule of *Todarodes pacificus* squamous hemocyanin (TpH) shows the complex oligosaccharide structures of the glycoprotein [8]. Chemical structure analysis of *Todarodes pacificus* hemocyanin revealed that the two most abundant sequences, HexNAcMan_3_GlcNAc_2_ and HexHexNAcMan_3_GlcNAc_2_, accounted for more than 95% of all oligosaccharides tested in all individuals. The Cu_2_O_2_ cluster and the carbohydrates of functional unit “d” of TpH are represented in the 3D-model of Figure 2. The authors demonstrated the influence of glycans on the dissociation and re-association behavior of the decameric forms of TpH [9].

Therefore, the great scientific interest in molluscan hemocyanins is due not only to their complex quaternary structures, but also to their great diversity in oligosaccharide structures, high carbohydrate content (2–9%), specific monosaccharide composition [10,11,12,13], and their well-established link with the immunostimulatory effect [14,15].

The studies on carbohydrate structures of hemocyanins from molluscs occupy an important part of this review, demonstrating significant differences in carbohydrate structures of molluscan hemocyanins.

## 2. Carbohydrate Structures of Hemocyanins

The first analyses of the carbohydrate structures of hemocyanins from the earth snail *Helix pomatia* (HpH) [10], freshwater snail *Lymnaea stagnalis* (LsH) [16], and the California mussel (keyhole limpet) *Megathura crenulata* (KLH) [13,17] present complex carbohydrate structures composed predominantly of the monosaccharides Xyl, Fuc, Man, Gal, GalNAc, and GlcNAc. Along with common monosaccharides, other monosaccharides such as Xyl and O-methyl-Gal have also been identified in HpH glycans [10,11,18]. Though the Xyl residue is less common in glycoproteins from animal organisms, their presence is essential for immunogenic effects [10].

Other specifically-modified structures have been identified that bind (β1-2)Xyl residues to β-Man and α-Fuc to 1,6-GlcNAc from the core of the glycan. Traces of large neutral N-glycans ending with 3-O-MeGal, Xyl, and/or Fuc monosaccharides have been found in *Planorbarius corneus, Achatina fulica,* and *Arion lusitanicus* [12,19]. The presented complete N-glycan spectrum of the slug *Arion lusitanicus* includes a combination of carbohydrate structures found in mammals, plants, insects, and the like [12,19]. Complex structures have also been published for some glycans of *Rapana thomasiana* hemocyanin (RtH), later renamed *Rapana venosa* (RvH) [11,20].

Furthermore, the studies on the carbohydrate structure of KLH, used in medicine, present two new *N*-linked glycans, Fuc(α1-3)GalNAc(β1-4)[Fuc(α1-3)]GlcNAc and Gal(β1-4)Gal(β1-4)Fuc(α1-6), as part of the huge protein molecule [13,17]. In one glycan, a Gal monosaccharide is linked to the final GalNAc, for which the clinical success of KLH in the treatment of bladder cancer is found. The carbohydrate chain Gal(β1-4)Gal(β1-4)Fuc(α1-6) is linked to the nucleus of another glycan, which also contributes to the use of KLH in immunotherapy.

The structural diversity of *N*-glycans in snails has been investigated very thoroughly by a number of authors who report predominantly high-mannose carbohydrate chains. Exceptions were found for some carbohydrate structures that bind sialic acid, α1-6- and α1-3-Fuc, β1-2-Xyl, MeMan, MeGal, GlcNAc, GalNAc, and other monosaccharides, but significantly fewer glycans end up with a 3-*O*-MeMan residue [10,11,18].

A great variety of oligosaccharide structures has also shown for side chains linked to carbohydrate structures of hemocyanins from the abdominal mollusc *Lymnaea stagnalis*, the California mussel KLH, and others [12,14,16,19,21]. Specific enzymes, responsible for the formation of these complex structures, have been found in various organs and tissues of *Limax maximus*, *Lymnaea stagnalis*, and other representatives of the molluscs [12].

Important information for the positions of glycosylated Asn residues for three FU-s (FU-b, FU-f and FU-h) is provided by the di-decameric model of KLH1 (Figure 1a–d). The model shows 120 potential *N*-glycan motifs (NXT and/or NXS), located mainly on the surface of the molecule (Figure 1d). FU-c is of special interest due to its absence of a residual glycan (Figure 1b) [22]. This corresponds to KLH-c with only one *O*-glycosylation site, lacking an attached, glycan moiety [23]. The absence of glycan in FU-c was also observed in hemocyanins of *Haliotis tuberculata* (HtH), *Octopus dofleini* (OdH), and *Aplysia californica* (AcH) [24]. Published literature mostly presents *O*-glycosylated hemocyanins of the arthropodan species, while very few *O*-linked glycans have been found in hemocyanins of the Molluscs. 

Another interesting feature for molluscan and arthropodan hemocyanins is the position of the putative glycosylation sites, located mainly on the surface of the molecule. Two *N*-linked sites of FU HtH1-h are located at rather unusual positions and are not found in other FUs, including HtH2-h. These glycans probably perform an important role related to the function of glycosylated FU-h, suggesting that they prevent binding of decamers to di-decamers and formation of multi-decamers in hemocyanins. The assumption is confirmed by the established absence of multi-decamers in the hemocyanins of AcH and HtH1 [24,25].

In this review, the oligosaccharide structures of three hemocyanins are presented: from the Black Sea snail *Rapana venosa,* the garden snail *Helix lucorum,* and the abalone *Haliotis tuberculata,* which normally exist under different living conditions. They have been compared with the well-studied oligosaccharide structure of keyhole limpet hemocyanin (KLH) from a mussel inhabiting the northern coast of America, and other hemocyanins with complex carbohydrate structures. 

Mass spectral methods and techniques are appropriate for analysis of glycoproteins and identification of their structure. To determine molecular masses of amino acid sequences (AAS) of proteins and peptides, mass spectrometry is used as the nucleus of proteomics. It is one of the most widely used methods for the analysis of biomolecules like peptides, proteins, and glycoproteins, with a series of sophisticated instrumentations such as Matrix Assisted Laser Desorption/Ionisation–Time of Flight (MALDI-TOF-TOF), Liquid Chromatography-Electrospray Ionization-Mass Spectrometry (LC/ESI-MS), Liquid Chromatography (LC-Q-trap-MS/MS) or Nano- Electrospray Ionization-Mass Spectrometry (nano-ESI-MS).

## 3. Carbohydrate Structure of Hemocyanins from the Marine Snail *Rapana venosa*

Recent research reported on the glycosylated nature of RvH, with a carbohydrate content of 8.9%, which differs for the two subunits RvH1 (12.4%) and RvH2 (4.4%) [26,27]. A new strategy for analysis and determination of the carbohydrate structure of subunits and FUs of RvH is presented here (Figure 3).

In the first approach, functional units (FUs) of the two subunits RvH1 and RvH2 are treated with trypsin and the obtained glycopeptides are identified through LC/ESI-MS, LC-Q-trap-MS/MS or nano-ESI-MS [28,29,30,31,32,33]. The second approach includes isolation of glycans after hydrolysis of the subunits of RvH with specific glycosidases (PNGase F) and analysis of the glycan structures with MALDI-TOF-MS, СЕ-MS, and Q-trap instrumental systems [28,29,34].

### 3.1. First Approach for Determination of the Carbohydrate Structure of RvH

The developed strategy is suitable for the determination of the complex carbohydrate structures of molluscan hemocyanins. The first approach of the developed and implemented strategy provides information about the oligosaccharide structures only for the RvH. 

#### 3.1.1. Isolation and Characterization of Glycopeptides of FUs from RvH

Three FUs (RvH1-a, RvH1-f, and RvH1-b) are isolated after trypsinolysis of RvH1 and the resulting fragments are separated on a Superdex 300 column. The positive orcinol/H_2_SO_4_ test allows one to detect two (Glp1 and Glp2) glycopeptides in RvH1-a, but only one (Glp3) in RvH1-f. Confirmation of the complex carbohydrate structures of the three glycopeptides (Glp1, Glp2, and Glp3) is presented after treatment with various glycosidases and analysis with CE-MS and ESI-MS. 

Detailed data for the structures of three glycopeptides from RvH1 (Glp1, Glp2, and Glp 3) are received after treatment with specific glycosidases and analysis of cleaved products with СЕ-MS and ESI-MS. That allows the identification of complex bi-antennary structures of the glycans including methylated galactoses bound to Glp1 and Glp2 differing from the structure of the glycopeptide Glp3 of RvH1-f [28,29].

With these published data on the carbohydrate structures of molluscan hemocyanins, methylated monosaccharides are mentioned, as found before, as constituents in seven related complex structures, constituents of the hemocyanins from *Lymnaea stagnalis* [16].

#### 3.1.2. Structures of Glycopeptides from RvH, Determined Through LC/ESI-MS, Nano-ESI-MS and Q-trap LC/MS/MS Systems

After trypsinolysis of the two subunits, RvH1 and RvH2, and separation of fragments on a Nucleosil 7 С18 column, mass spectrometric methods were applied for structure determination of glycopeptides [28]. The presented orcinol /H_2_SO_4_ test (Figure 4) gives positive reactions only for 58 eluted fractions from the column. The oligosaccharide structures of the glycopeptides were determined through tandem-MS after tracking of the fragmented ions from the MS/MS data of the double-charged ion [M+2H]^2+^ and scanning of the pseudo-MS/MS/MS.

The summary of the results obtained from µLC/ESI-MS analysis after tryptic hydrolysis of RvH1 provide additional confirmation for the glycosylation of RvH1: three glycopeptides were identified with Mw 2745.5 Dа, 2421.0 Dа, and 2907.1 Dа, and oligosaccharide structures of the high-mannose type [26,35]. Data for the carbohydrate structure of RvH was completed with the comprehensive nano-ESI-МS analysis of isolated RvH1 and RvH2 glycopeptides after trypsinolysis. This methodology gives simultaneous access to fundamental structural characteristics of the glycoproteins, such as their glycan structure, amino acid sequences (AAS), and binding sites of glycans to the peptide chains [31]. The structure of the glycopeptides are determined after the interpretation of the Y- and B-ions represented in the spectrum, giving information about the calculated glycopeptide, with Mw 2676.32 Dа, and ion, with *m*/*z* 1338.16 [M+2H]^2+^ (Figure 5). Another important feature from the MS/MS spectrum is the bound glycan (GlcNAc residue) at the ion at *m*/*z* 1661.71 to the *N*-glycosylated peptide (R-) at *m*/*z* 1459.75 [M+H]^+^. The glycan-typical oxon ions [M+H]^+^ with *m*/*z* 204.08 (HexNAc) and *m*/*z* 366.15 (HexHexNAc) in the MS/MS spectrum show the exact structure of the glycan and the bound peptide. The glycopeptide characteristic is complemented by the represented different single fragment [M+H]^+^ glycan ions with *m*/*z* 528.20 (Hex_2_HexNAc), *m*/*z* 690.27 (Hex_3_HexNAc), *m*/*z* 852.31 (Hex_4_HexNAc), and *m*/*z* 1014.46 (Hex_5_HexNAc) (Figure 5). 

The results of the nano-ESI-MS analysis of the glycopeptide with Mw 2676.3 Da confirm the highly mannose-like carbohydrate structure of RvH [31]. The assumption of the carbohydrate structure of RvH1 was confirmed by the combined analysis of glycopeptides with LC/MS and the Q-trap system. The resulting LC/MS/MS chromatogram of the eluted fraction at 31.24 min at high scanning resolution expressed a dominant ion with *m*/*z* 837.97 [M+3H]^3+^, corresponding to a glycopeptide with Mw 2511.91 Da. The fragment Y- and B-ions with *m*/*z* 204.1, 366.2, 528.3, 690.5, and 852.4 determine the carbohydrate structure as Hex_2_Man_2_GlcNAc_2_ of a glycan with Mw 1054.0 Da. The other fragmented y- and b-ions reflect the amino acid sequence AAS of the peptide MGQYGD (I/L) STNNTR with two potential glycosylation sites (–D(L/I) S- and –NNT-) (Figure 6). According to LC/MS and Q-trap analysis, only one glycosylation site (–NNT) in a glycopeptide with Mw 2511.91 Da, to which a high mannose glycan Hex_2_Man_2_GlcNAc_2_ with Mw 1054.0 Da is linked, can be detected [31].

### 3.2. Second Approach for Determination of Carbohydrate Structure of RvH 

Interpretation of the structure of glycopeptides is often difficult due to the overloaded MS spectra via fragmented ions of both the peptide (y- and b-ions) and the glycan (Y- and B-ions). This disadvantage is solved by a second approach. After treatment with the specific glycosidase PNGase, F glycans are removed from the subunits RvH1 and RvH2 [32,33], eluted with 2 mL 25% AСN/ 0.05% TFA through a Carbograph column and finally analysed by СЕ and MALDI-TOF, presenting the high mannose character of glycan structures including Gal, Fuс and Xyl in the carbohydrate chains of RvH1 and RvH2 [32].

#### 3.2.1. Analysis of N-Glycans from RvH by Capillary Electrophoresis Combined with MS Analysis

The approach of CE with MS allows structure identification of glycans in very low concentration mixtures. Moreover, using this method, a new type of structure was presented for the ion at *m*/*z* 555.7 [M-3H]^3−^ of the glycan from RvH1, which was different from structures published by other authors. Giles et al. 2005 published one МеHex residue connected to an inner Fuc residue (Figure 7A, B) [33].

The analyses of CE-MS/MS show the complex structures of the glycans of RvH1 presented by ions at *m*/*z* 555.7 [M-3H]^3−^ and two isomeric forms of the glycan with identical MS/MS spectra. The identical MS/MS spectra demonstrate essential divergences in the structure of glycan in connection to the GlcNAc residue at α-1,3 or α-1,6-positions in the trisaccharide of the glycan. Moreover, this structure resembles the published structure from Gielens and co-authors with an internal GalNAc residue and MeHex [11] (Figure 7A). 

The used methods cannot differentiate the two structures of MeHex and HexA because of the small difference in the Mw from 0.036 Dа. This problem was solved by CE-MS/MS analyses, which determined the methylated structure of the oligosaccharide, whose behavior does not correspond to the glycan at *m*/*z* 555.7. Larger glycans will migrate faster than smaller ones, and charged oligosaccharides will migrate slower than their neutral counterparts. A glycan, detected at *m*/*z* 555.7 as a 4-fold negatively charged ion, migrates slower than glycan, which is smaller and detected at *m*/*z* 557.8 as 3 times negatively charged ion.

Based on this, we can conclude that the ion at *m*/*z* 555.7 corresponds to a charged N-glycan and a complex glycan with an internal fucose connecting a hexuronic acid (HexA) and an N-acetylhexosamine (HexNAc). Similar structures with HexA and Fuc residues are suggested from MS/MS analyses and the observed electrophoretic offset of the three ions [M-4H]^4−^ in RvH. The specific structures with HexA in RvH1 are most possibly a result of the oxidizing environment [33].

#### 3.2.2. Analysis of Glycans from RvH by 3-AP- and APTS-Labeled Glycans

А new combined approach of CE-MS and MS/MS analyses of labeled glycans of RvH1 with positively charged 3-aminopyrazole (3-AP) is suitable to determine glycan structures at very low concentrations in mixtures. A comparative analysis of the behaviour of the 3-AP-labeled glycans shows a shift of the elution time in the chromatogram [33].

Another important piece of information is the structural differences of glycans with identical isomer MS/MS spectra, which is probably due to the acid tetrasaccharide attached to the α-1,3 or α-1,6-arm positions of the trimannose core. Proof for this is the reported difference from the additional Fuc residue in the MS/MS spectra of the ion with *m*/*z* 853.7 [M+2H]^2+^ (Figure 8A) and with *m*/*z* 926.8 [M+2H]^2+^ (Figure 8B).

Confirmation of the position of the triple moiety (HexNAcFucHexA) is ion B_2_ with *m*/*z* 526.1 [M+H]^+^, indicating the loss of the Fuc residue after separation from the HexNAc and HexA residue. The advantage of the applied approaches is the identification of the complex structures of two glycans FucHexAHexNAcGlcNAcMan_3_GlcNAc_2_ and Fuc_2_HexAHexNAcGlcNAc_2_Man_3_GlcNAc_2,_ respectively.

#### 3.2.3. Determination of Hexuronic Acid in the Structure of RvH1 After Amidation and Permethylation of the Glycans

This new method is based on the modification of the carboxyl groups of the glycan with NH_4_Cl and DMT-MM [4-(4,6-dimethoxy-1,3,5-triazin-2yl)-4-methyl-morpholinium chloride], whereby the carboxylic acid is converted to an amide, leading to a decrease in the Mw of acidic glycans by 0.9840 Dа, identifying a bound HexA moiety of the glycan [33]. MALDI-MS analysis reported a decrease in Mw for eight N-linked glycans in RvH1 after amidation, demonstrating the involvement of eight HexA in glycans of RvH1. The conversion of HexA to MeHex shows an increase in Mw by 14 Da after permethylation of the four identified glycans, noted in Table 1 as 14, 17, 19 and 21 [32,33]. 

Another important method for determination of complex structures of the glycans in RvH is Q-trap analyses of permethylated glycans. The presented MS/MS spectra of structures with tetrasaccharide are expressed by two ions with *m*/*z* 660.3 [M+Na]^+^(B_3_/Y_6β_) and 919.4 [M+Na]^+^(B_3_) (Figure 9). The position of the Fuc residue in the chain was confirmed by additional MS^3^ analysis of the permethylated B_3_ ion. The reported change in Mw by 160 Da [145+14] probably reflects methylation of only one hydroxyl group of the glycan because the other three hydroxyl groups participate in other bonds. Evidence of acidic carbohydrate structures is also presented in the RvH2 subunit. They are reported based on the change in molecular masses with 1 D of two glycans from the glycan mixture of RvH2 before and after amidation [32]. 

Further interesting carbohydrate structures are also presented in RvH1 and RvH2 by Q-trap analyses of the obtained glycans after treatment of both subunits with PNGase F glycosidase, specific for N-glycosylation. This approach proved to be very suitable for determination of the structures of 25 glycans from RvH1 and 28 glycans from RvH2. Glycan structures with one HexNAc residue and HexA bound in an oligosaccharide of RvH to an internal Fuc residue have been demonstrated [32,33]. 

#### 3.2.4. Structures of the N-Glycans of RvH1 and RvH2 Determined by the Q-trap System 

The comparison of the data obtained from the various employed methods and techniques determines the complex carbohydrate structure of *Rapana venosa* hemocyanin, similar to that observed for most molluscan hemocyanins. With the new methods developed, rare carbohydrate moieties of glycans, such as MeHex residues, have been detected. For the first time, a new class of glycans with an α1,6-Fuc residue bound to HexA and HexNAc has been discovered for hemocyanins. 

## 4. Carbohydrate Structure of Hemocyanins from *Helix lucorum*

New methods and approaches for analyzing the carbohydrate structure of hemocyanin from the snail *Helix lucorum* (HlH) show а complex carbohydrate structure (Figure 10). The glycans obtained after treatment of βc-HlH with PNGase F are determined by MALDI-MS analyses, mainly as triple charged [M-3H]^3+^ ions [36,37,38]. The determined methylated monosaccharides in glycan structures may be MeMan or MeGal found in βc-HlH. It is important to note the absence of acid glycans with HexA at βc-HlH (Figure 11) [38].

The obtained results from conducted experiments of APTS-labeled *N*-glycans and CE-MS also confirm the complex structures of HlH [37]. Similar structures have been published for other molluscan hemocyanins, such as *Helix pomatia, Lymnaea stagnalis* [16], and *Arion lusitanicus* [19], suggesting that methylated *N*-glycans perform an important function in the body. 

Important information is presented on the complex structure of some glycopeptides, such as the designated Xyl moiety attached to the Man residue from inside the glycan, as well as the *N*-linked glycosylation site to which the glycan is attached. 

The position of the glycans in the hemocyanin is also expressed after mass spectral analysis of some βc-HlH glycopeptides. An important advantage of analyzing the carbohydrate structure of HlH is its determined nucleotide sequence, which helps determining the motifs “NXT” and “NXS”, and glycosylation of the sites accordingly.

Based on this information, 13 *N*-binding sites in βc-HlH, 14 in αD-HlH, and 8 in αN-HlH are identified on different positions in their FUs. Only one potential site is located in βc-HlH-e, two in βc-HlH-a, -f, and -g, and three *N*-binding sites in βc-HlH-d and -h (Figure 11) [39]. 

## 5. Carbohydrate Structure of Hemocyanins from *Haliotis tuberculata*

The complex carbohydrate structure of subunit HtH1 from molluscan hemocyanin *Haliotis tuberculata* (HtH) is also presented. Preliminary data from MALDI-MS, performed before and after treatment of the glycans from subunit HtH1 with the specific exoglycosidases β1-3,4,6-galactosidase and α-1,6- and α-2,3,4-fucosidase, confirm the complex structure of the glycoprotein. After application of the second approach, 15 new structures of *N*-linked glycans to HtH1 were determined by Q-trap tandem mass spectrometry [40].

One complex structure of HtH1 is presented in Figure 12 with a glycan at *m*/*z* 1002.8 [M+2Na]^2+^ and structure of MeHex_2_HexMan_3_GlcNAc4Fuc_2_, including two Fuc and two terminal MeHex residues bound to internal GlcNAc.

The large set of ions in the MS/MS spectrum determines the structure of the glycan, the positions of two Fuc residues in the said chains, and one bound Fuc(α1-6) residue to the pentasaccharide core in the glycan. On the basis of the obtained data, a conclusion about the wide variety of structures of the 15 *N*-linked glycans of HtH1 was made (Figure 13). 

As a general feature of the structures of glycans 1, 5, 8, 10, 12, 13, 14, and 15, a bound Fuc (α1-6) residue to the pentasaccharide nucleus is outlined. Another interesting feature is the structures of glycans 4, 12 and 15 of HtH1 with MeHex, which is also found in other hemocyanins of the molluscan species.

The advantage of the new methods and approaches developed should be noted, which allows the determination of complex structures of glycans 6, 8, 9, 10, 12, 13, 15, and 16 in which a novel MeMan [Fuc(α1-3)-]GlcNAc structural motif is linked to an internal GlcNAc residue.

The discovered modifications in glycan 7 are interesting and complex, where a Xyl residue is attached to carbon-2 (C2) of Man from the core Man_3_GlcNAc_2_ or a Fuc residue is attached to C6 of GlcNAc. Chemical structure analysis of the oligosaccharides of *Todarodes pacificus* hemocyanin revealed that the two most abundant sequences HexNAc+Man_3_GlcNAc_2_ and HexHex+NAcMan_3_GlcNAc_2_ accounted for >95% of oligosaccharides in all individuals tested [9].

## 6. Glycosylated Sites of Hemocyanins from Molluscs

A general characteristic of the hemocyanins is the potential glycosylation sites in the molecule. The published nucleotide sequences can help us to suggest the features of the carbohydrate structures of the HlH, KLH, and HtH1 subunits [25,39].

### 6.1. Glycosylation Sites in Helix lucorum Hemocyanins

The established positions on the active sites in HlH are different for the αD-HlH subunit, where αD-HlH-c has one site, two are found in αD-HlH-d and-e, and three in FUs αD-HlH-а, -g, and -h. Partial information about potential binding sites is presented for the αN-HlH subunit, because of the indefinite full amino acid sequences, with one site located in FUs αN-HlH-a,-b,-g, and -h and two in αN-HlH-d and αN-HlH-e. 

In confirmation of the data on the location of the putative sites on the surface of the molecule, the positions of the glycans in the constructed 3D-model of FU βc-HlH-g are presented [41] (Figure 14). The presented 3D-model of FU βc-HlH-g is composed of two domains, the active site and one putative *N*-linked glycosylation site at positions 125–127, which are located in/on one of the “central” domains. Another site is located at position 372–374 in the “β-sandwich” domain with both putative sites exposed to the surface of the molecule. They are mainly concentrated between β11- and β12-sheet structures, as reported for the FUs βс-HlH-a, -d, -e and -g [39]. Different positions have important roles in the oligomerization of the molecule and the formation of di-decameric forms. *N*-glycosylation sites located at the C-end of FU “h” from αD-HlH and βc-HlH also perform a similar function. As a general characteristic of the three subunits of HlH, a potential glycosylation site in the loop between the α16-helical and β11-sheet structures of the β-sandwich domain can be noted. Another feature of almost all molluscan hemocyanins is the absence of a potential *N*-linked glycosylation site in FU “c”. This information is confirmed for both βc-HlH and αN-HlH subunits with *N*- glycosylation sites also not found for FU-s βс-HlH-b, αD-HlH-b, αD-HlH-f, and αN-HlH-f.

A similar motif, -Asn-Pro-Thr-, is found in FUs βc-HlH-g and αN-HlH-g, but glycan binding is blocked by a Pro residue. 

This conclusion is valid for all molluscan hemocyanins, but not for the FU αD-HlH with one putative glycosylation site at position 331–333 (-AsnAspThr-). This site corresponds to the site of position 143 (-AsnThrSer-), determined by the nucleotide sequence of RvH-c, which was further demonstrated by the orcinol/H_2_SO_4_ assay [31]. The location of two glycans on the surface of the 3D-model of βc-HlH-g bonded to the glycosylation sites Asn125 and Asn245 corresponds to the published crystallographic structures of the Odh-g and RtH2-e FUs [42,43].

Тhe nucleotide sequences of HtH1 and HtH2 subunits are also known [24,44], and were applied to construct 3D-models of eight FUs in which glycans are located on the surface of the molecule (Figure 15). Two models deserve special attention: one is FU HtH1-c, in which a potential glycosylation site is absent, and HtH1-h, where one of the two presented glycans is exposed on the surface of FU HtH1-h [40]. 

Based on the presented data, an important role for this glycan has been suggested, which probably impedes the re-association of dissociated HtH1 subunits that do not fibrillate into di-decamers or larger aggregates as observed in some molluscan hemocyanin carbohydrate structures. The new information on the structure of hemocyanin from the garden snail *Helix lucorum* and *Haliotis tuberculata* is that they have an area of glycoproteins with predominantly methylated carbohydrate structures, which do not include acidic glycans.

The full characteristics of the complex oligosaccharide structure of hemocyanins from Molluscs are presented after identifying the connecting sites for the glycans to the polypeptide chains of RvH. The absence of information for the amino acid sequence (AAS) of RvH makes the determination of the glycosylation sites difficult. However, this is presented after comparative analysis of drawn results from different applied methods and approaches:Edman degradation of glycopeptides;Analysis of the peptides after de-glycosylation with specific glycosidases; Separation of glycopeptides on the columns and determination of AAS after MS/MS analyses. 

*N*- linked glycosylation sites have been determined in isolated glycopeptides from FUs and subunits of RvH1 using a new method, including labelling of the *N*-linking sites, followed by treatment of the glycopeptides with PNGase F enzyme in buffer containing 50% H_2_^18^O [26,29,33].

### 6.2. Glycosylation Sites in RvH 

#### 6.2.1. Glycosylation Sites in RvH Determined by nano-ESI-MS and Q-trap-LC/MS/MS Systems 

The information provided on the glycosylated structure of RvH has also been confirmed after the determination of glycosylation sites with the development of a new method of analysis of the separated glycans from the polypeptide chain to which the first *N*-linked HexNAc residue remains. An important advantage of the method is fragmentation of the peptide chain by nano-ESI-MS and Q-trap-LC/MS/MS systems, which takes place in the absence of glycans in the cuvette [33]. 

There are two possible glycosylation sites in the eluted 31.24 min glycopeptide with AAS MGQYGNXSTNNTR. After triple quadrupole scanning, only one site was shown to be *N*-glycosylated (-NXS-). A drawback of the method is the insufficient information to demonstrate the glycosylation sites of all tested glycopeptides. This problem is solved after applying a new approach that involves MS/MS analyses of nano-LC-ESI-MS/MS triple charged ^18^O-labeled peptides and proof of the glycosylation site.

#### 6.2.2. Glycosylation Sites in RvH Determined After ^18^O-Labelling of Peptides

An additional characteristic of the glycosylation sites in RvH is presented by applying a new approach after separation of the *N*-glycans from the protein by glycosidase PNGase F and marking of the glycosylation sites of RvH1 in buffer with 50% H_2_O. The glycosylation sites show a difference of 2 Da between *m/z* of the ^18^O-labeled and unlabeled ions of the MS/MS spectra [33].

This new method is very sensitive and suitable for the analysis and the recognition of glycosylation sites in other glycoproteins, but the presented limited amount of data from the MS/MS spectrum does not allow for *de novo* analysis to determine the AAS of glycopeptides. This is achieved by a nano-LC-ESI-Q-trap system where, at slow fragmentation over time (TОF) of ^18^O-labeled peptides, very low ions are detected in the MS spectra (Figure 16). 

The y-ions and the labeled b3 with *m*/*z* 316.1050 [M+H]^+^ in the TDF-MS spectrum of the ^18^O-marked peptide with *m/z* 966.0 [M-2H]^2+^ AAS with one *N*-linked glycosylation site -DLT- was determined [33]. Results from the advantage of the chosen method for detecting ^18^O-labeled glycopeptides in biological solution are presented in Table 2. Preliminary ion scanning at 204 *m*/*z* of the 23-min elution fraction showed the most intensive ion with *m*/*z* 973 [M+2H]^2+^, which represented a glycopeptide with Mw 1943.6 Da ([972.3 × 2] − 1 = 1943.6) (Figure 17A,B) [33].

Analysis of the ion with *m*/*z* 905.2 [M+H]^+^ (a peptide without glycan, Figure 17C) from the MS/MS spectrum proves that one glycan FucMan_3_GlcNAc_2_ is attached to the (-NI/LT-) site of the glycopeptide (Table 2). Mass spectra report a difference of 1 Da between the values obtained for a peptide after glycan release with PNGase F and those of ^18^O-labeled ions. The difference reported is evidence of a glycosylation site represented by a group of [M+H]^+^ ions resulting from the sequential fragmentation of the bound peptide with GlcNAc [33].

#### 6.2.3. Glycosylation Sites in RvH, Determined After Fragmentation of the Genome 

The positions of glycosylation sites are also demonstrated by the nucleotide sequences of hemocyanin of *Haliotis tuberculata*, *Aplysia californica*, *Nucula nucleus* and *Nautilus pompilius* [7,22,24,25,44,45], and two fragments of the RvH genome that partially encode FU-b and FU-c [31]. The negative Orcinol/H_2_SO_4_ test provides information on the absence of an *N*-linked site in RvH-b, confirmed by the obtained 316 amino acid residues (AARs) of RvH-b, as the positive test for sugars corresponds to one *N*-linked site (-Asp-Thr-Ser-) at the 143rd position identified in the nucleotide sequence of RvH-c.

Data obtained from MS/MS analysis of the glycopeptide with *m*/*z* 1099.19 [M-2H]^2+^ (Table 2, № 10) for site AAS (FSGEVDGHNTSR) were corrected by nucleotide sequence. It is known that the Gly residue is not always reflected in the MS/MS spectrum. This was confirmed by the nucleotide analysis of the glycopeptide, where the Trp residue is replaced by -GlyGlu-. The information obtained is consistent with the published putative *N*-linked glycosylation sites for the FUs HtH2-c, OdH-c, and NpH-c (Figure 18) and serves as a basis for determination of other RvH characteristics.

The results of the constructed 3D-model of RvH-c represent the located glycans on the surface of the molecule (Figure 19) [31], which is confirmed by the crystallographic structure of hemocyanin FUs of *Todarodes pacificus* (Figure 1) [9]. The location of *N*-linked glycosylation sites in molluscan hemocyanins is most commonly observed between β11- and β12-sheet structures in the β-sandwich domain of the FUs [25]. For the OdH-d and NpH-d FUs, these positions are similar, but for OdH-g and NpH-g they differ, which is related to the influence of the formed *N*-glycan site of the opposite FU OdH-g. One site in NpH-g located at the same position, but with no bound glycan, has been demonstrated [22].

The crystal structure of *Todarodes pacificus* hemocyanin revealed *N*-linked glycosylation at five of seven possible sites, including Asn387 for FU-a, Asn806 for FU-b, Asn1498 and Asn1636 for FU-d, and Asn2472 for FU-e. Four of these sites are located at identical positions, however, FU-d possesses an additional glycan (Figure 20) [9].

The oligosaccharides were all located on the surface of the wall and formed carbohydrate clusters. The carbohydrates conjugated at Asn387 and Asn806 of protomer A (hereafter, glycan-387 A and glycan-806 A), and formed a cluster with glycan 1498 which was located at the interface among three protomers, i.e., protomers A, I, and J. Furthermore, this site is located at the interface between two plate-like protomer-dimers.

Molluscan hemocyanins are oligomeric glycoproteins with complex dodecameric quaternary structures and heterogeneous glycosylation patterns, primarily consisting of mannose-rich *N*-glycans, which contribute to their structural stability and immunomodulatory properties in mammals [46]. The enzyme-catalyzed *N*-deglycosylation of *Concholepas concholepas* [47], *Fissurella latimarginata* [2], and KLH disrupts their quaternary structure and impairs their immunogenic effects. Biochemical analyses confirm the importance of glycans by showing that deglycosylation does not alter the secondary structure of hemocyanin, but alters their re-folding mechanism and dodecameric structure.

## 7. Conclusions and Future Direction

The data provide new information on the carbohydrate structures of oxygen-binding glycoproteins, ‘hemocyanins’, with two copper atoms in the active site determined by a combination of new methods and techniques. Complex carbohydrate structures have a fucosylated pentsaccharide core and a high degree of methylation.

It is important to note that, along with the common structures, all molluscan hemocyanins stand out with specific glycan structures. Of great interest are the first discovered acid glycans in RvH1 and RvH2, where at two positions from the internal Fuc residue, HexNAc and HexA is bound. The novel MeHex [Fuc (α1-3)] GlcNAc structural motif has also been demonstrated in HtH1 glycans, which is composed of one or two MeHex and (α1-3) Fuc bound to internal GlcNAc and an additional Xyl residue.

One of the main characteristics of hemocyanins from the snails *Helix lucorum, Helix pomatia, Lymnaea stagnalis,* and *Arion lusitanicus* is the proven predominantly methylated glycan structures of 3-*O*-methyl-d-mannose and 3-*O*-methyl-d-galactose. Similar structures have been published for other molluscan hemocyanins, suggesting that methylated *N*-glycans perform an important function in the body.

Comparative analyses of the primary structures of HtH1, βc-HlH, and RvH present a high degree of homology to the putative *N*-linked glycosylation sites located on the surface of the FUs. The position of the glycans on the surface of the molecule was also proven by the constructed 3D models of βc-HlH-g, RvH-c, as well as the eight FUs of HtH1, which correspond to the represented positions of the crystallographic structures of OdH-g and RtH2-e [42,43].

The obtained data from these studies provide new information on the carbohydrate structures of molluscan hemocyanins combining typical structural characteristics of various higher organisms (mammals, plants, insects, nematodes and trematodes). This explains the scientists’ interest in these glycoproteins, which are a suitable target for determining the effect of glycans on the structure, properties and function of glycoproteins.

## Figures and Tables

**Figure 1 biomolecules-10-01470-f001:**
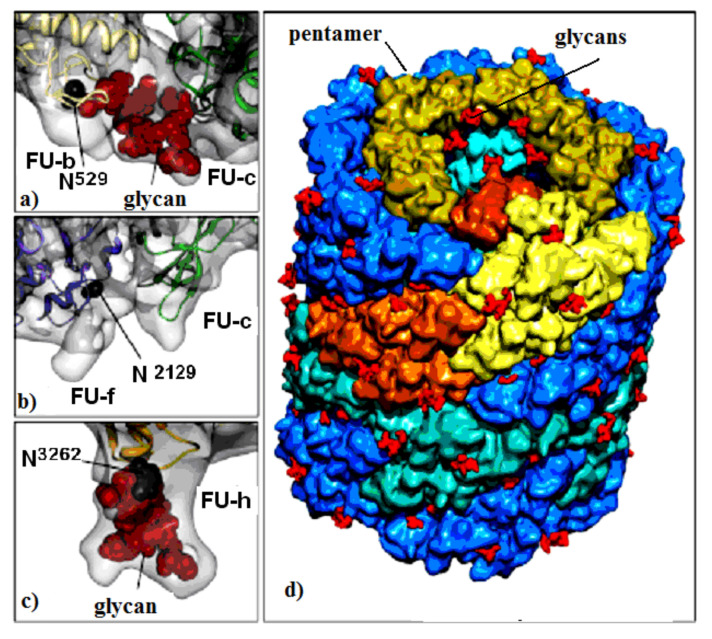
KLH1 subunit model. Location of Asp residue in the glycosylation site of: (**a**) FU-b and FU-с; (**b**) FU-f and FU-с; (**c**) FU-h; (**d**) di-decamer KLH1 model and the position of 120 glycans (marked in red) [7].

**Figure 2 biomolecules-10-01470-f002:**
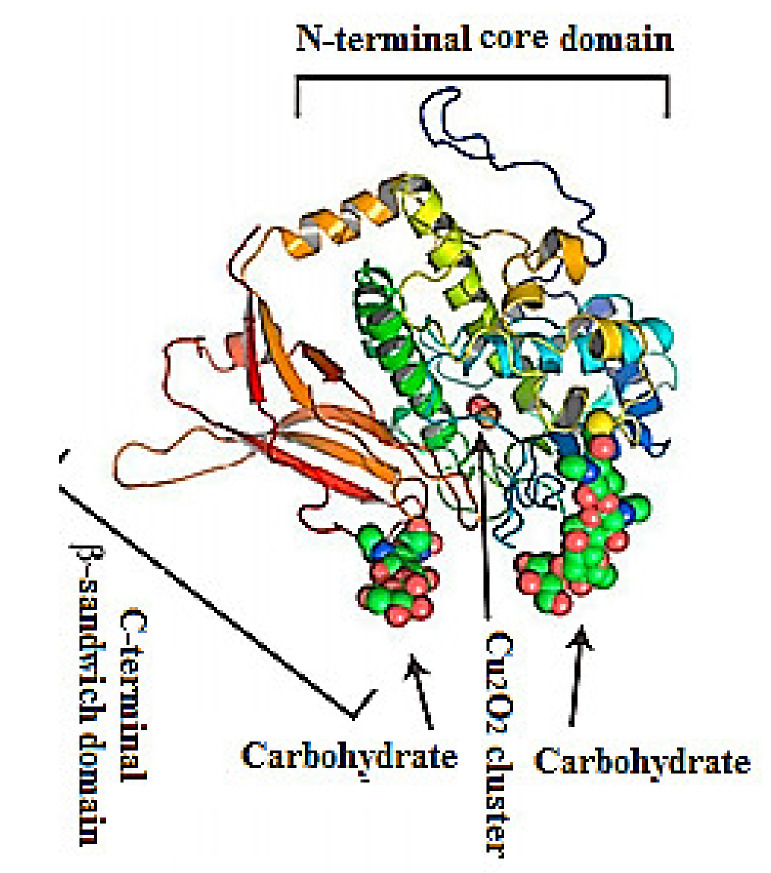
Representative ribbon diagram of FU-d of *Todarodes pacificus* squamous hemocyanin, colored as a ramp from blue (N-terminus) to red (C-terminus). The Cu_2_O_2_ cluster and carbohydrates [9].

**Figure 3 biomolecules-10-01470-f003:**
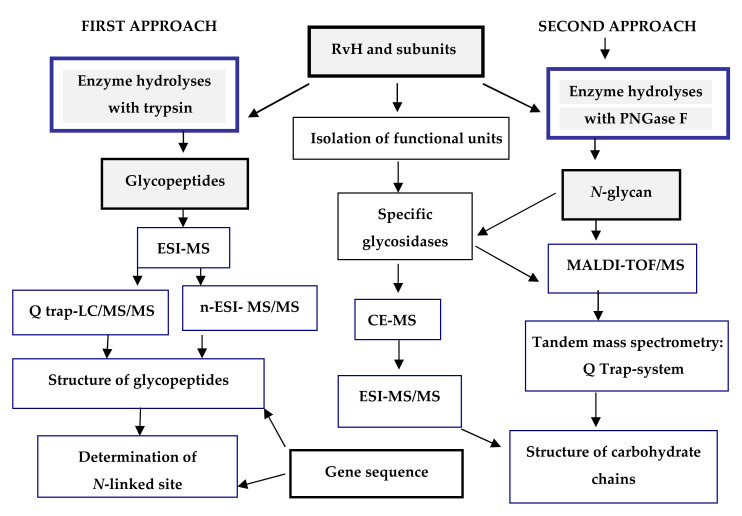
Scheme for analysis and determination of the carbohydrate structure of RvH1, RvH2 and FUs. On the basis of this strategy, two main approaches are involved.

**Figure 4 biomolecules-10-01470-f004:**
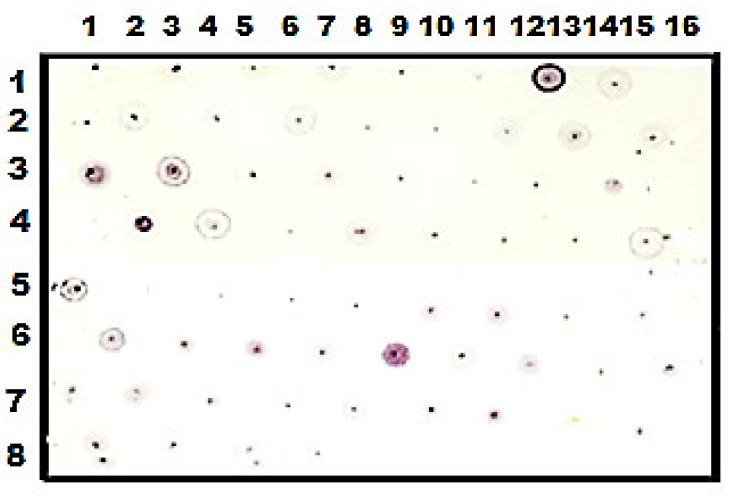
Orcinol /H_2_SO_4_ test of the obtained fractions of RvH1 after trypsinolysis.

**Figure 5 biomolecules-10-01470-f005:**
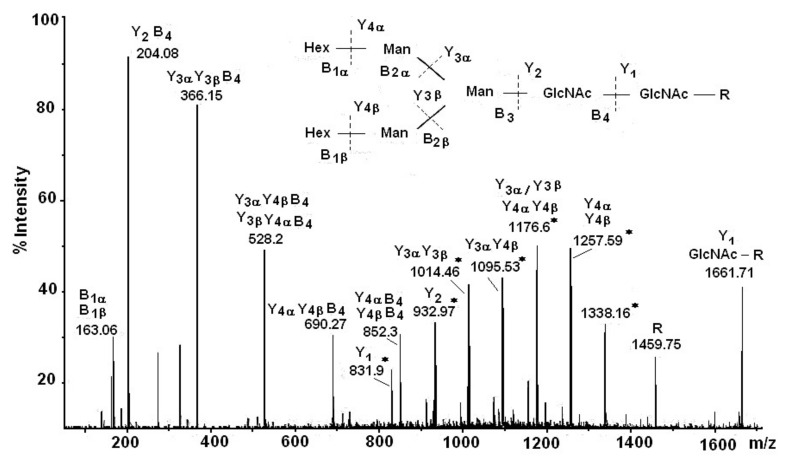
Nano-ESI- MS/MS analysis of the glycan G1, bound to glycopeptides, presented as an ion at *m*/*z* 1338.16 [M+2H]^2+^ [31].

**Figure 6 biomolecules-10-01470-f006:**
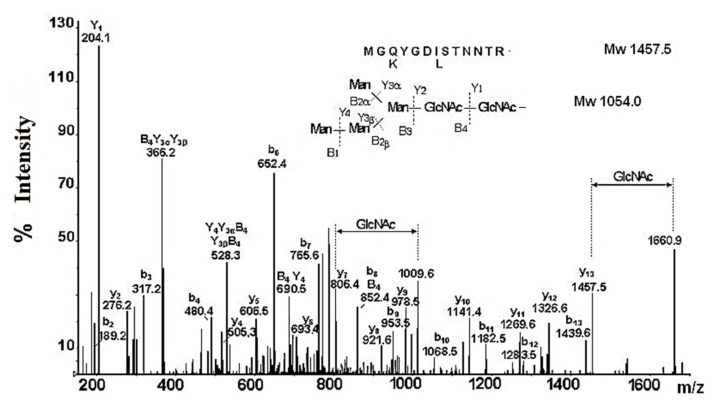
Enhanced product ion (EPI)—scanning of an ion with *m*/*z* 837.97 [M+3H]^3+^ with a higher resolution.

**Figure 7 biomolecules-10-01470-f007:**
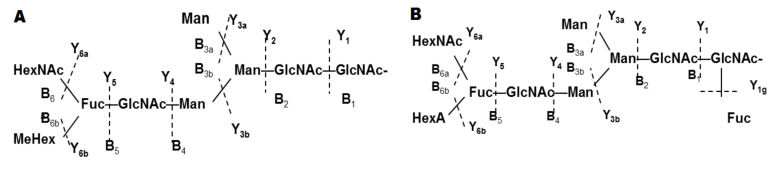
The complex structures presented for the ion at *m*/*z* 555.7 [M-3H]^3−^ of RvH1 analyzed by CE-MS/MS and MS/MS spectra: (**A**) Structure presented by Giles et al. 2005 [11]; (**B**) Structure presented by Sandra et al. 2007 [33].

**Figure 8 biomolecules-10-01470-f008:**
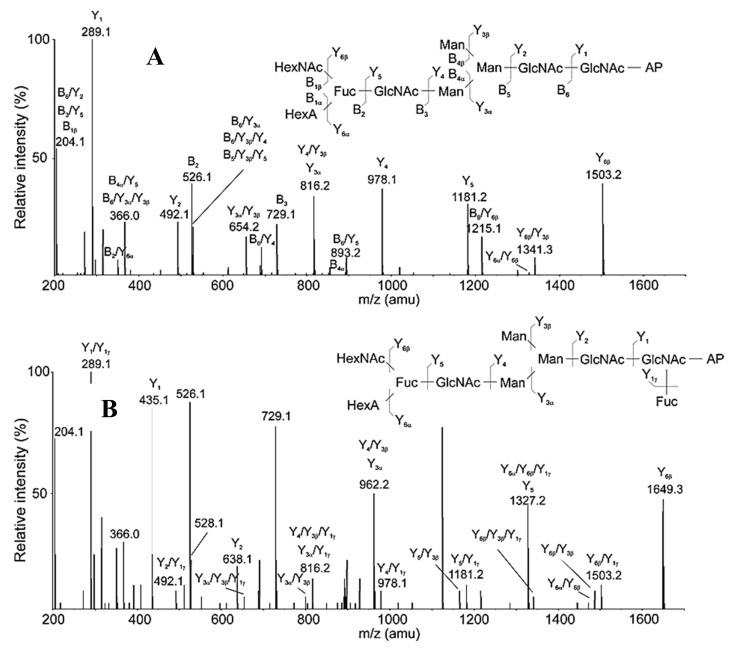
(**А**). MS/MS analysis of 3-AP-glycan from RvH with *m*/*z* 853.7 [M+2H]^2+^ eluted at 14.87 min. (**B**) MS/MS analysis of 3-AP-glycan from RvH with *m*/*z* 926.8 [M+2H]^2+^, eluting at 14.87 min.

**Figure 9 biomolecules-10-01470-f009:**
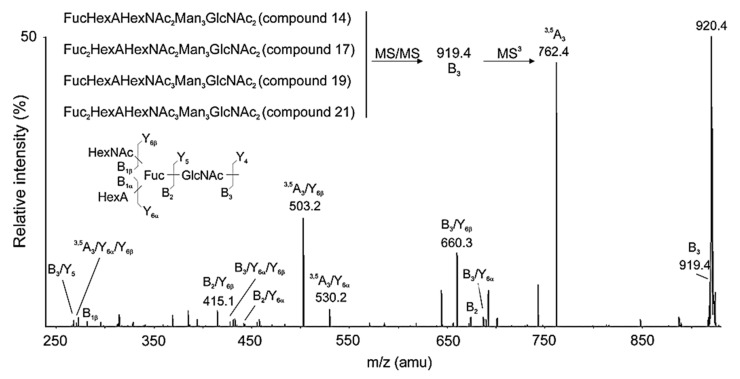
MS^3^ spectrum of the B_3_ ion with *m*/*z* 919.4 [M+Na]^+^ (tetrasaccharide), after methylation of acid glycan from RvH1 with *m*/*z* 1038.5 [M+2Na]^2+^ (structural information including permethylated sites).

**Figure 10 biomolecules-10-01470-f010:**
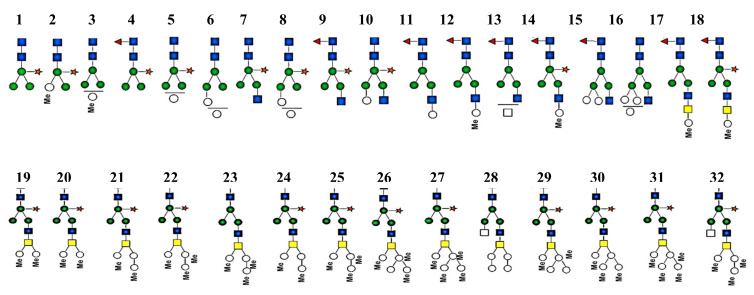
Neutral *N*-attached glycans from βc-НlН, obtained after treatment with PNGase. Monosaccharides, presented through nomenclature of the consortium of functional glycomics.

**Figure 11 biomolecules-10-01470-f011:**
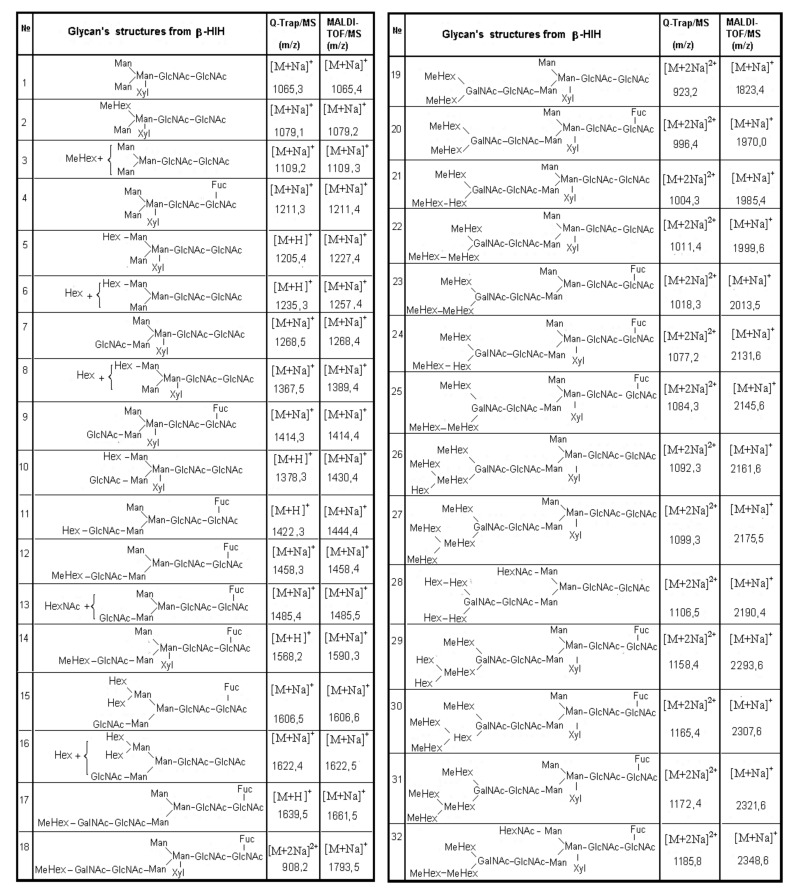
Structures of glycans from β-HlH, determined through different methods [37].

**Figure 12 biomolecules-10-01470-f012:**
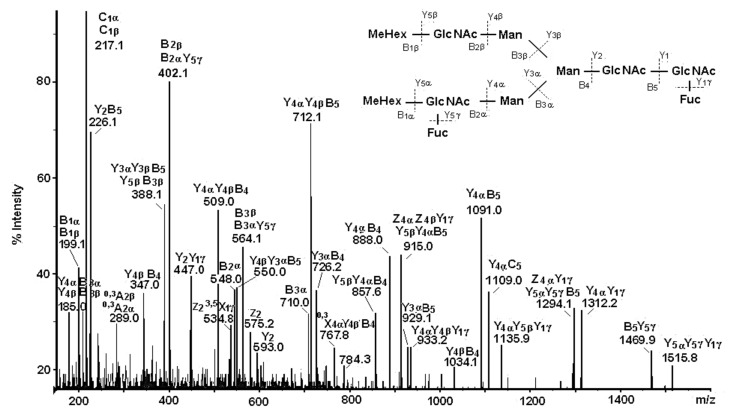
MS/MS spectrum of HtH1 glycan represented by ion with *m*/*z* 1002.8 [M+2Na]^2+^ and complex carbohydrate structure [40].

**Figure 13 biomolecules-10-01470-f013:**
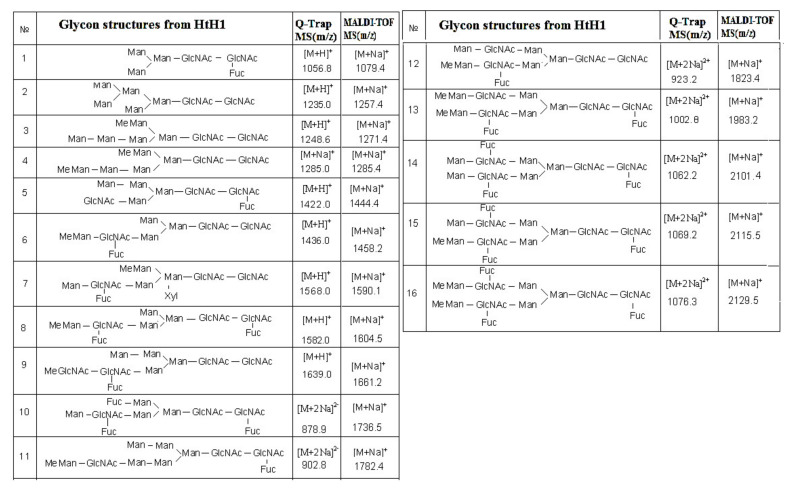
Structure of the glycans isolated from HtH1

**Figure 14 biomolecules-10-01470-f014:**
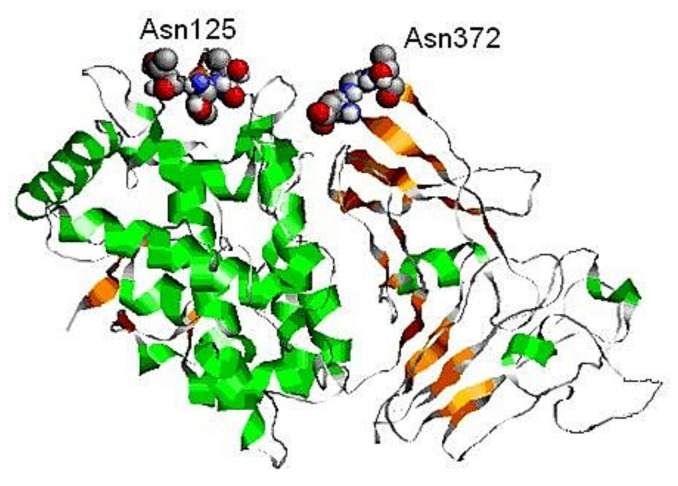
3D-model of FU βc-HlH-g, built through the Swiss PDB viewer program and compared with the *Octopus dofleini* (OdH-g) FU model “g” [41]. Two possible glycosylation sites, to Asn125 and Asn245 residue, are indicated [41].

**Figure 15 biomolecules-10-01470-f015:**
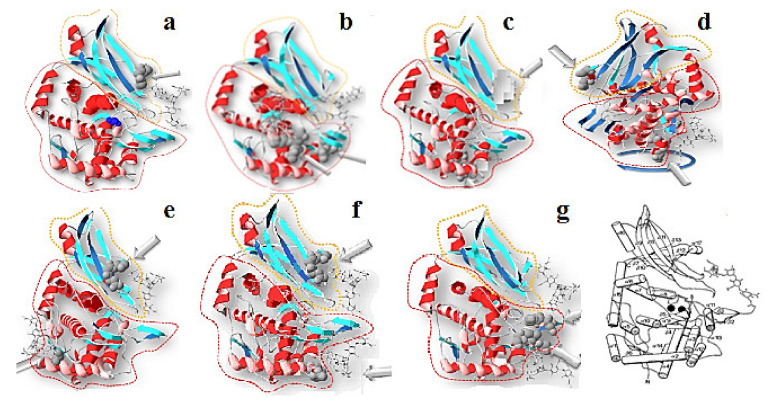
3D-models of FU from «a“ to «g“ of HtH1 and potential *N*-linked sites, depicted on the basis of the structure of OdH-g. β-sandwich domain (yellow); α-nucleus (red); *N*-glycolised site (grey). The glycans are marked with arrows [40].

**Figure 16 biomolecules-10-01470-f016:**
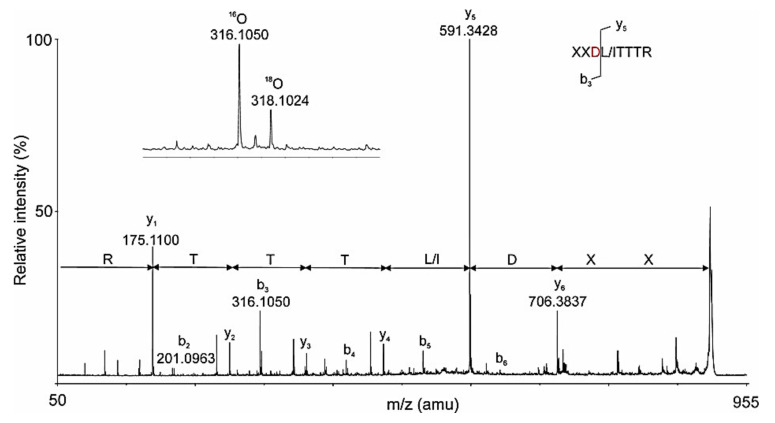
TОF spectrum of ^18^O-labeled peptide, presented from an ion with *m*/*z* 966 [M-2H]^2+^; collision energy 28 eV and TDF time 8 ms. В) The labeled ion b_3_ with *m*/*z* 316.1050 is depicted on the figure.

**Figure 17 biomolecules-10-01470-f017:**
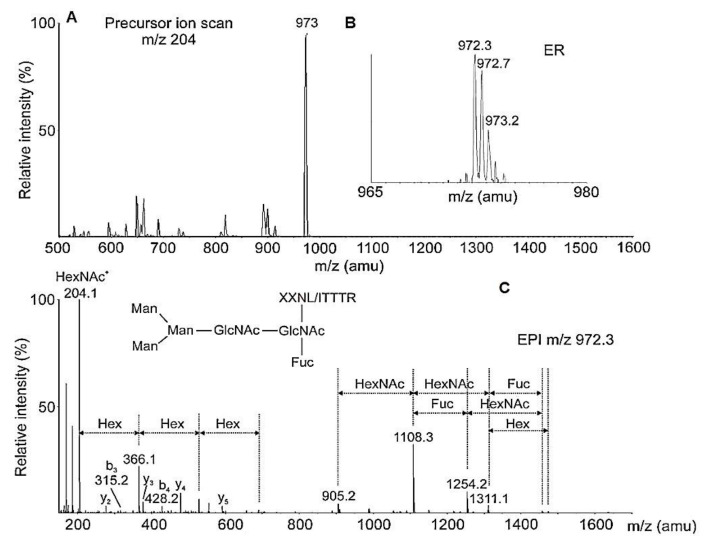
(**A**) Ion scanning of 23-min eluted fraction. (**B**) ER scanning of the intensive ion; (**C**) CEPI scanning of fragmenting ions. Increase of the mass of ion b3 with 1 Da in EPI spectra is compared to the MS/MS of the labeled peptides [33].

**Figure 18 biomolecules-10-01470-f018:**
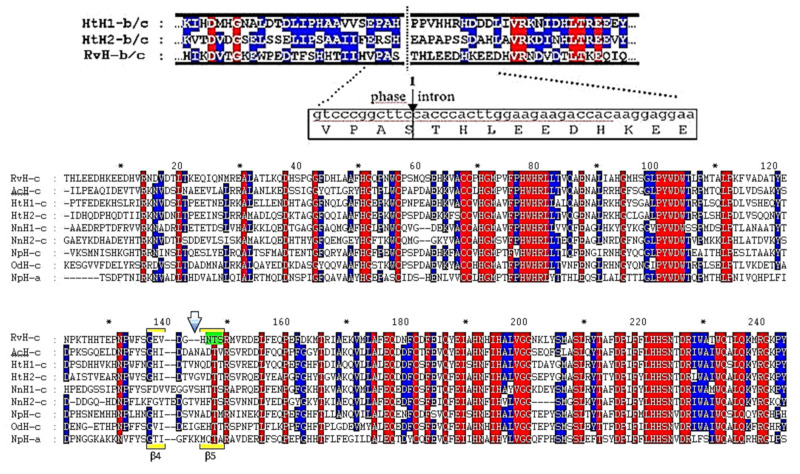
Comparative analysis of the AAS of RvH-c with FUs from other hemocyanins. Conservative sites of the AAS (in blue and red) and *N*-linked glycosylation site of RvH-c (in green) [31].

**Figure 19 biomolecules-10-01470-f019:**
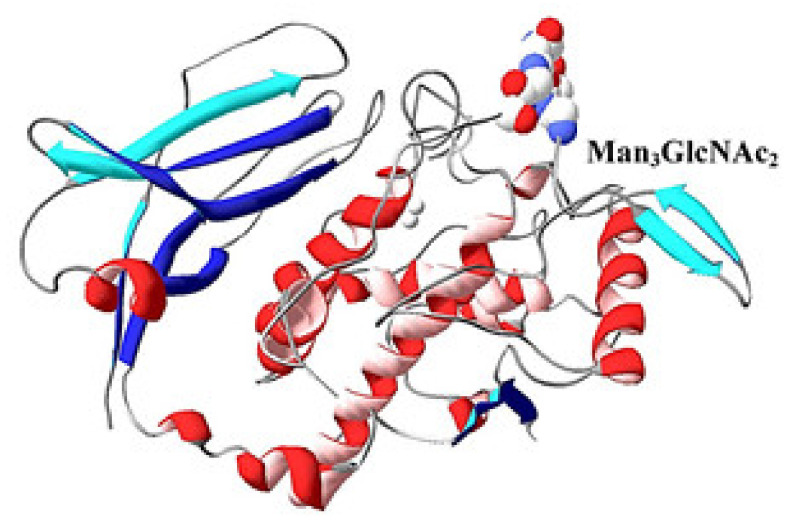
3D-model of FU RvH-c, depicted on the basis of the structure of the hemocyanin from *Octopus dofleini* [41]. Glycan, attached to the polypeptide chain of FU-c [31].

**Figure 20 biomolecules-10-01470-f020:**
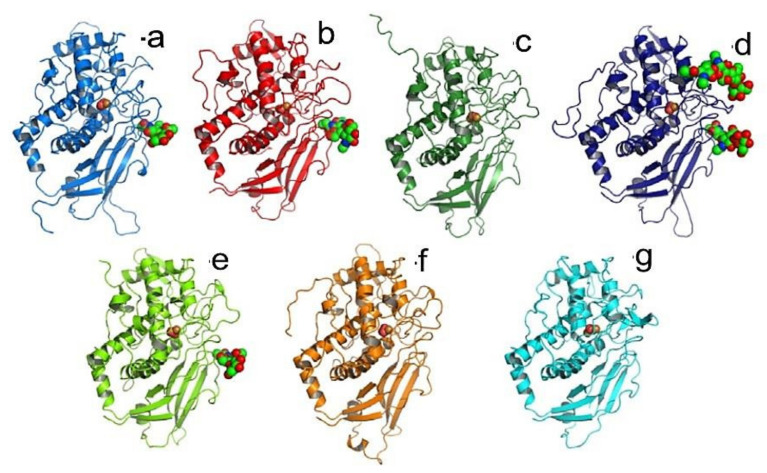
*N*-linked glycosylation sites of functional units of *Todarodes pacificus* hemocyanin [9].

**Table 1 biomolecules-10-01470-t001:** Composition of N-glycans in RvH1. (**a**) N-glycans are represented as [M+Na]^+^ ions. Mw of permethylated glycans is indicated in parentheses. (**b**) *m*/*z* values of APTS-labeled glycans [33].

№	Structure	MALDI [M-H]^+^	№	Structure	MALDI [M−H]^+^
1	FucMan2GlcNAc2	917.3	12	Man7GlcNAc2	1581.7
2	Man3GlcNAc2	933.3	13	Fuc2HexNAc2Man3GlcNAc2	1631.8
3	FucMan3GlcNAc2	1079.5	14	FucHexAHexNAc2Man3GlcNAc2	1661.7
4	Man4GlcNAc2	1095.5	15	FucHexNAc3Man3GlcNAc2	1688.8
5	HexNAcMan3GlcNAc2	1136.5	16	Man8GlcNAc2	1743.7
6	FucMan4GlcNAc2	1241.6	17	Fuc2HexAHexNAc2Man3GlcNAc2	1807.8
7	Man5GlcNAc2	1257.6	18	Fuc2HexNAc3Man3GlcNAc2	1834.
8	FucHexNAcMan3GlcNAc2	1282.6	19	FucHexAHexNAc3Man3GlcNAc2	1864.8
9	HexNAc2Man3GlcNAc2	1339.6	20	Man9GlcNAc2	1905.8
10	Man6GlcNAc2	1419.6	21	Fuc2HexAHexNAc3Man3GlcNAc2	2010.8
11	FucHexNAc2Man3GlcNAc2	1485.7			-

**Table 2 biomolecules-10-01470-t002:** AAS and carbohydrate structures of glycopeptides from RvH analyzed by ESI-MS, Q-Trap-LC/MS/MS and Q-Trap system. X is unknown IF [28,29,33].

Glycopeptides		Glycans	Mass/Charge
(*m*/*z*)			
*CE*			
1	XVYSV*N*GTLLGAHVLGSR	Man3GlcNAc2 HexA HexNAc2 Fuc2 Man3GlcNAc2	941 [M+3H]^3+^ 1233 [M+3H]^3+^
2	X—XFSWVDGH***N***TSR	GlcNAcMan3GlcNAc2 FucMan3GlcNAc2 Man3GlcNAc2	1168 [M+3H]^3+^ 1149 [M+3H]^3+^ 1104 [M+3H]^3+^
3	FQ***N***DTSLDGYQAVAEFHGLPAK	FucMan3GlcNAc2	1153 [M+3H]^3+^
4	FQ***N***DTSLDGFQAVAEFHGLPPK	GlcNAcMan3GlcNAc2	1010 [M+4H]^4+^
5	LHSYSGSYL***N***ASLLHX—X	Man3GlcNAc2	968 [M+2H]^2+^
6	X***N***GTELSPR / X***N***ASELSPR	Hex5Man3GlcNAc2	1373 [M+2H]^2+^
***ESI-MS***			
7	[QK]AE***N***LTTTR	FucMan3GlcNAc2	1036 [M+2H]^2+^
8	AE***N***LTTTR	Fuc Man3GlcNAc2	972 [M+2H]^2+^
9	HHGHV[...K...***N***...]R	Fuc Man3GlcNAc2	1396 [M+2H]^2+^
10	FSWVDGH***N***TSR	Man3GlcNAc2	1099 [M+2H]^2+^
11	YE[IL]HAV***N***GST[IL]AA[IL]	Hex3Man3GlcNAc2	1339 [M+2H]^2+^
12	YE[IL]HAV***N***GST[IL]AA[IL]	Hex3Man3GlcNAc2	1419 [M+2H]^2+^
***Q trap-LC/MS/MS***			
13	MGQYGDLST***N***NTR	Hex Man3HexNAc2	837.9[M+2H]^3+^
14	SV***N***GTLLGSQILGKPY	Fuc Man3GlcNAc2	896 [M+3H]^3+^
15	FSWVDGH***N***TSR	Man3GlcNAc2	1099 [M+2H]^2+^
16	AE***N***ITTTR	Fuc Man3GlcNAc2	972 [M+2H]^2+^
17	FA***N***ATSIDGPNA	SO4 MeHexAMeHexNAc2Man3GlcNAc2	2786 [M+H]^+^
18	EMLTL***N***GTNLA	MeHex2AHexNAc2Man3GlcNAc2	2846 [M+H]^+^
19	IHSYSGSYI***N***ASLLHGPSII	MeManMan2GlcNAc2	2848 [M+H]^+^

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
