# Peer review of "De Novo Structural Determination of the Oligosaccharide Structure of Hemocyanins from Molluscs"

_biomolecules, 2020, doi:10.3390/biom10111470_

Round 1

Reviewer 1 Report

This manuscript described the carbohydrate structure of hemocyanins from MOLLUSCS. They report that the variation of the determination of glycans and glycopeptides performed with the most common used methods for the analysis of biomolecules, including peptides and proteins like MALDI-TOF-TOF (time of flight), LC/ESI-MS, LC-Q-trap-MS/MS nano-ESI-MS and others. The review is worth to publish. However, the manuscript needs to be revised before publication.

The comments:

1. The title of the manuscript is too short. Authors had better change the title with more descriptive way.

2, Most of abbreviations should be rechecked, such as KLH1 (line 46), GlcNAc (not GlcNac line59, GlcNA line 75), and Mw (not Mm, line 172).

3. The composition and m/z are not matched, such as line 189 of Hex5HexNAc2 and line 200 of Hex2Man3GlcNAc2.

4. There was wrong explanation of composition and Figure (line 204).

5. These are no A and B in Figure 7, line 232. What’s the meaning of 229 in line 230?

6. These are no A and B in Figure 8, line 248.

7. Please check the number of line 263, five identified glycans, this is four.

8. Please insert the number of 22 in line 263.

9. Table 1 in line 268 is not clear, for example, is this for RvH or RbH1? There are no (a) and (b). MALDI means [M-H]+ or [M+Na]+?

10. In line 272 of Figure 9 legend, this is difficult to understand that the number of Table or permethylated [M+2Na]2+?. If the number indicated the Table, it should be changed with structural information including permethylated sites.

11. In line 290, [M-3H]3+ ion are unclear, more explain it.

12. Please check the line 331, as Table 2 not Table 1.

13. In line 464, these is no C in Figure.

Author Response

Dear Editor and reviewers,

We wish to thank you all for your constructive comments in this round of review. Your comments provided valuable insights to refine its contents and analysis. In this document, we try to address the issues raised as best as possible.

Comments and Suggestions for Authors

  1. The title of the manuscript is too short. Authors had better change the title with more descriptive way.

Answer : The title of the manuscript was corrected ”De Novo Structural Determination of Oligosaccharide Structure of Hemocyanins from Molluscs“

2, Most of abbreviations should be rechecked, such as KLH1 (line 46), GlcNAc (not GlcNac line59, GlcNA line 75), and Mw (not Mm, line 172).

Answer : All remarks are corrected in the text.

  1. The composition and m/z are not matched, such as line 189 of and line 200 of Hex2Man3GlcNAc2.

Answer : m/z 1338.16 [M+2H]2+ is a glycopeptide G1 not Hex5HexNAc2 structure and was corrected.

m/z Hex2Man3GlcNAc2 was corrected to Hex2Man2GlcNAc2

  1. There was wrong explanation of composition and Figure (line 204).

Answer : Composition is corrected to Hex2Man2GlcNAc2

  1. These are no A and B in Figure 7, line 232. What’s the meaning of 229 in line 230?

Answer : Figure 7 was corrected and A and B were added.

  1. These are no A and B in Figure 8, line 248.

Answer : Figure 8 was corrected and A and B were added.

  1. Please check the number of line 263, five identified glycans, this is four.

Answer : four identified glycans noted in Table 1 as 14, 17, 19 and 21

  1. Please insert the number of 22 in line 263.

Answer : Number 22 was removed from the table.

  1. Table 1 in line 268 is not clear, for example, is this for RvH or RbH1? There are no (a) and (b). MALDI means [M-H]+ or [M+Na]+?

Answer : In Table 1 RvH1 is presented and the ions are [M+Na]+

  1. In line 272 of Figure 9 legend, this is difficult to understand that the number of Table or permethylated [M+2Na]2+?. If the number indicated the Table, it should be changed with structural information including permethylated sites.

Answer : Text was changed with structural information including permethylated sites

  1. In line 290, [M-3H]3+ ion are unclear, more explain it.

Answer : Text was corrected to be more clear “The glycans obtained after treatment of βc-HlH with PNGase F are presented on MALDI-MS analyzes mainly as triple charged [M-3H]3+ ions [36-38]. ”

  1. Please check the line 331, as Table 2 not Table 1.
  2. In line 464, these is no C in Figure.

Answer : Text was corrected

Reviewer 2 Report

     The Authors carried out a deep review on the characterization of the glycan structures of molluscan  hemocyanins, with basis on more than forty publications on this topic. The most important aspect here underlined is that glycosylation is extyremely important for the biological and therapeutic activity of these glycoproteins. The topic is quite complicated, therefore we are especially asking to improve the language as a whole, carrying out an extensive revision.

1) Here we indicate some of the sentences that need to be rewritten because, besides the language and english problem, even the meaning is often unclear:

lines 206-207; 218-220; 225-228;277-279; 281-282;330-331; 474-476 and possibly others.

2) Please verify also the correctness of the following words:

-biantenni instead of biantennary

-tetrasacchiride instead of tetrasaccharide

-N-glycosylated center instead of N-glycosylation site

-consorcium instead of consortium

-specter instead of spectrum

-glycon structures instead of glycan structures

3) Some figures, composed of A), B), C) indicate these letters only in the legend and not on the figure. In Fig. 15 it is hard to identify what could be Fig. 15 C

4) The chapter on Conclusions should be numbered 7 and not 6 and Table 4 is repeatedly named Table 3

5) On lines 426 and 443 are we not dealing with H218O?

6) Line 331: we could not find the cited information in Table 1 of this manuscript.

Author Response

The Authors carried out a deep review on the characterization of the glycan structures of molluscan  hemocyanins, with basis on more than forty publications on this topic. The most important aspect here underlined is that glycosylation is extremely important for the biological and therapeutic activity of these glycoproteins. The topic is quite complicated, therefore we are especially asking to improve the language as a whole, carrying out an extensive revision.

  • Here we indicate some of the sentences that need to be rewritten because, besides the language and english problem, even the meaning is often unclear:

lines 206-207; 218-220; 225-228;277-279; 281-282;330-331; 474-476 and possibly others.

Answer :

206-207 – text Due to the overloaded MS spectra via fragmented ions of the peptide (y- and b-ions) and the glycan moieties (Y- and B-ions) of the spectra is often impeded. was replaced “Interpretation of the structure of glycopeptides is often dificult due to the overloaded MS spectra via fragmented ions of both, the peptide (y- and b-ions) and the glycan (Y- and B-ions).” 

218-220 – The complex structures presented for the ion at m/z 555.7 [M-3H]3– of RvH1 and analyzed by CE-MS/MS determine two isomeric forms of the glycan with identical MS/MS spectra and propose one and the same carbohydrate structures. This sentence was replaced by :

The analyzes of CE-MS MS show the complex structures of the glycans of RvH1 presented by ions at m/z 555.7 [M-3H]3–and two isomeric forms of the glycan with identical MS / MS spectra and proposing one and same carbohydrate structure.

225-228 -  This problem was solved through CE-MS/MS analyses for the identification of the methylated structure of the oligosaccharide, which behaviour does not correspond to a charged glycan and the HexA in the carbohydrate structure of the glycan is suggested [33]. This sentence was replaced by :

This problem was solved by CE-MS / MS analyzes to determine the methylated structure of the oligosaccharide, whose behavior does not correspond to the glycan at m/z 555.7. Larger glycan will migrate faster than smaller ones, and charged oligosaccharides will migrate slower than their neutral counterparts. Detected glycan at m/z 555.7 as a 4-fold negatively charged ion migrates slower than smaller glycan, detected at m/z 557.8 as a 3 times negatively charged ion. Based on this are conclude that the ion at m/z 555.7 corresponds to a charged N-glycan and a complex glycan with an internal fucose connecting a hexuronic acid (HexA) and an N-acetylhexosamine (HexNAc)

277-279 - Evidence of acidic carbohydrate structures in RvH2 is also reported by the change of the molecular masses of two glycans with 1 Da for each glycan after amidation of the glycan mixture of the subunit [32]. This sentence was replaced by :

Evidence of acidic carbohydrate structures is also presented in RvH2 subunit. They are reported based on the change in molecular masses with 1 D of two glycans from the glycan mixture of RvH2 before and after amidation [32].

281-282 - Another interesting carbohydrate structures are also presented after RvH1 and RvH2 treatment respectively with PNGase F glycosidase and Q-trap analysis of the obtained glycans. This sentence was replaced by :

Another interesting carbohydrate structures are also presented  in RvH1 and RvH2 by Q-trap analyses of the obtained glycans after treatment of both subunits with PNGase F glycosidase.

330-331 –  Based on this information, 13 N-binding sites in βc-HlH, 14 in αD-HlH, and 8 in αN-HlH are identified on different positions in FUs, with the only one potential center located in βc-HlH-e, two in βc-HlH-a, -f, and –g, and three centers in βc-HlH-d and -h (Table 1). This sentence was replaced by :

Based on this information, 13 N-binding sites in βc-HlH, 14 in αD-HlH, and 8 in αN-HlH are identified on different positions in their FUs. Only one potential center is located in βc-HlH-e, two in βc-HlH-a, -f, and –g, and three centers in βc-HlH-d and –h (Table 2)  [40].

474-476 – The obtained 316 amino acid residues (AARs) of RvH-b provide important information confirming the orcinol/H2SO4 assay for the absence of an N-linked site, while the incorporated N-linked site (-Asp-Thr-Ser-) at 143-rd position in the nucleotide sequence of RvH-c corresponds to the positive test for sugars. This sentence was replaced by :

The negative Orcinol/H2SO4 test provides information for the absence of an N-linked site in RvH-b, confirmed by  the obtained 316 amino acid residues (AARs) of RvH-b as the positive test for sugars corresponds to one N-linked site (-Asp-Thr-Ser-) at 143-rd position identified in the nucleotide sequence of RvH-c.

2) Please verify also the correctness of the following words:

-biantenni instead of biantennary

-tetrasacchiride instead of tetrasaccharide

-N-glycosylated center instead of N-glycosylation site

-consorcium instead of consortium

-specter instead of spectrum

-glycon structures instead of glycan structures

Answer : - corrected in the text and figures

3) Some figures, composed of A), B), C) indicate these letters only in the legend and not on the figure. In Fig. 15 it is hard to identify what could be Fig. 15 C

Answer : A), B), C) indications are included

4) The chapter on Conclusions should be numbered 7 and not 6 and Table 4 is repeatedly named Table 3

Answer : text is corrected

5) On lines 426 and 443 are we not dealing with H218O?

Answer : 18O-labeled peptides with H218O

6) Line 331: we could not find the cited information in Table 1 of this manuscript.

Text was corrected

Round 2

Reviewer 2 Report

The authors practically answered satisfactorily to all points.

There are some minor problems concerning the english language that can be resolved at the level of the journal

In chapter 6  the mentioned buffer contains 50% H2O or 50% H218O?

Author Response

The authors practically answered satisfactorily to all points.  The comments:

  1. There are some minor problems concerning the english language that can be resolved at the level of the journal.

Answer : Text was corrected

2, In chapter 6  the mentioned buffer contains 50% H2O or 50% H218O?

Answer : Text was corrected - line 440  - containing 50% H218O